# Infrared spectroscopic study of hydrogen bonding topologies in the smallest ice cube

Gang Li [1,5], Yang-Yang Zhang[2,5], Qinming Li[1,3,5], Chong Wang[1,3], Yong Yu[1,3], Bingbing Zhang[1], Han-Shi Hu[2], Weiqing Zhang[1], Dongxu Dai[1], Guorong Wu [1], Dong H. Zhang [1], Jun Li [2,4✉], Xueming Yang [1,4✉] & Ling Jiang [1✉]

The water octamer with its cubic structure consisting of six four-membered rings presents an excellent cluster system for unraveling the cooperative interactions driven by subtle changes in the hydrogen-bonding topology. Despite prediction of many distinct structures, it has not been possible to extract the structural information encoded in their vibrational spectra because this requires size-selectivity of the neutral clusters with sufficient resolution to identify the contributions of the different isomeric forms. Here we report the size-specific infrared spectra of the isolated cold, neutral water octamer using a scheme based on threshold photoionization using a tunable vacuum ultraviolet free electron laser. A plethora of sharp vibrational bands features are observed. Theoretical analysis of these patterns reveals the coexistence of five cubic isomers, including two with chirality. The relative energies of these structures are found to reflect topology-dependent, delocalized multi-center hydrogen-bonding interactions. These results demonstrate that even with a common structural motif, the degree of cooperativity among the hydrogen-bonding network creates a hierarchy of distinct species. The implications of these results on possible metastable forms of ice are speculated.

[1] State Key Laboratory of Molecular Reaction Dynamics, Dalian Institute of Chemical Physics, Chinese Academy of Sciences, 116023 Dalian, China. [2] Key Laboratory of Organic Optoelectronics & Molecular Engineering of the Ministry of Education, Department of Chemistry, Tsinghua University, 100084 Beijing, China. [3] University of Chinese Academy of Sciences, 19A Yuquan Road, 100049 Beijing, China. [4] Department of Chemistry, Southern University of Science and Technology, 518055 Shenzhen, China. [5] These authors contributed equally: Gang Li, Yang-Yang Zhang, Qinming Li. ✉email: junli@mail.tsinghua.edu.cn; xmyang@dicp.ac.cn; ljiang@dicp.ac.cn

As the most vital matter on the earth, water and its inter-action with other substances are essential in the life of our planet. Understanding the structure of bulk water and its hydrogen-bonding networks, however, remains a grand chal-lenge[1,2]. Spectroscopic investigation of water clusters provides a quantitative description of hydrogen-bond motions that are relevant to those in ice and liquid water[3,4]. Currently, cationic or anionic forms of water clusters have been extensively investigated because of relative ease in size-selection and detection[5–9]. These studies have provided essential knowledge on the structure and dynamics of the ionic water clusters.

Inasmuch as hydrogen-bonding networks in neutral water clusters are substantially different from those in ionic ones, to investigate neutral water clusters is a prerequisite to gain funda-mental insights into the structures and properties of ice and liquid water. Previous experimental and theoretical studies demon-strated that the water trimer, tetramer, and pentamer all have cyclic minimum-energy structures with all oxygen atoms in a two-dimensional (2D) plane, while the hexamer and heptamer have three-dimensional (3D) noncyclic structures[10–18]. We recently show that a 3D nonclyclic pentamer can coexist with its cyclic minimum-energy structure at finite temperature[15]. Among the small size clusters, of particular interest is the water octamer, which was proposed to represent the transition to cubic structures dominated in larger systems and display behavior characteristic of a solid ↔ liquid phase transition[19–22]. The low-energy structures of the water octamer were predicted to be nominally cubic[19–22], with the eight tri-coordinated water molecules taking up positions at the corners of the cube. Such tri-coordinated water molecules have been identified at the surface of ice[23–26]. The hydrogen bonds within the mostly crystalline subsurface layer are found to be stretched by the interaction with the disordered component[25]. The water octamer has thus become a superb benchmark for accurate quantification of the hydrogen-bonding interactions that govern the surface and bulk properties of ice.

Experimental characterization of the water octamer has been awkward due to the difficulty in size-selection and detection of neutral water clusters in general. Only a few gas-phase studies have been achieved[27–31], and two nearly isoenergetic structures with $D_{2d}$ and $S_4$ symmetry are found. Here we report the well-resolved infrared (IR) spectra of confinement-free, neutral water octamer based on threshold photoionization using a tunable vacuum ultra-violet free electron laser (VUV-FEL). Distinct features observed in the spectra identify additional cubic isomers with $C_2$ and $C_i$ sym-metry, which coexist with the global-minimum $D_{2d}$ and $S_4$ isomers at finite temperature of the experiment. Analysis of the electronic structure reveals a remarkable stability of these cubic water octa-mers arising from extensively delocalized multi-center hydrogen-bonding interaction.

## Results and discussion

**IR spectra of the water octamer.** The vibrational spectra were obtained using a VUV-FEL-based IR spectroscopy apparatus described in detail in "Methods" section[32]. In the experiment, neutral water clusters were generated by supersonic expansions of water vapor seeded in helium using a high-pressure pulsed valve (Even-Lavie valve, EL-7-2011-HT-HRR) that is capable of pro-ducing very cold molecular beam conditions[33]. For the IR exci-tation of neutral water clusters, we used a tunable IR optical parametric oscillator/optical parametric amplifier (OPO/OPA) system (LaserVision). Subsequent photoionization was carried out with about 30 ns delay with a VUV-FEL light at 113.30 nm delivered by the Dalian Coherent Light Source (DCLS) facility. IR spectra were recorded in the difference mode of operation (IR laser on–IR laser off).

The experimental IR spectrum of $(H_2O)_8$ in the OH stretching region is shown in the bottom of Fig. 1 and the band positions are listed in Supplementary Table 1. The comparison of present and previously measured spectra is given in Supplementary Fig. 1. From Supplementary Fig. 1, the present spectrum displays three distinct absorptions at 2980, 3002, and 3378 cm$^{-1}$; the 3460 cm$^{-1}$ band is now observed with high intensity, which was not observed in the helium-scattering IR spectrum[27] and only appeared with low intensity in the IR-UV spectra of benzene-tagged $(H_2O)_8$ (ref. [28]). Strikingly, the OH stretch spectra in the 3516–3628 cm$^{-1}$ region include many absorptions spanning multiple vibrational bands, which are considerably more complex than the spectra contributed by high-symmetry $D_{2d}$ and $S_4$ cubic octamers[27,28], suggesting the presence of more low-symmetry minima of the water octamer.

**Assignment of IR spectra of the water octamer.** To assign and analyze the observed spectral features, global-minimum structural search based on density functional theory (DFT) was accom-plished for the water octamer using TGMin code[34,35] (see theo-retical details in the "Methods" section), which lead to the location of totally 2784 distinct structures. Quantum chemical calculations were carried out to refine the energies of the low-lying isomers (within 11 kcal/mol) (Supplementary Fig. 2) using the ab initio MP2/aug-cc-pVDZ (AVDZ) method. The five lowest-energy structures for the water octamer (isomers **I**–**V**) are shown in Fig. 2. Each isomer has two classes of hydrogen-bonding environments that we classify as AAD and ADD con-figurations according to the number of acceptor (A) and donor (D) hydrogen bonds, respectively. The **I**–**V** structures differ

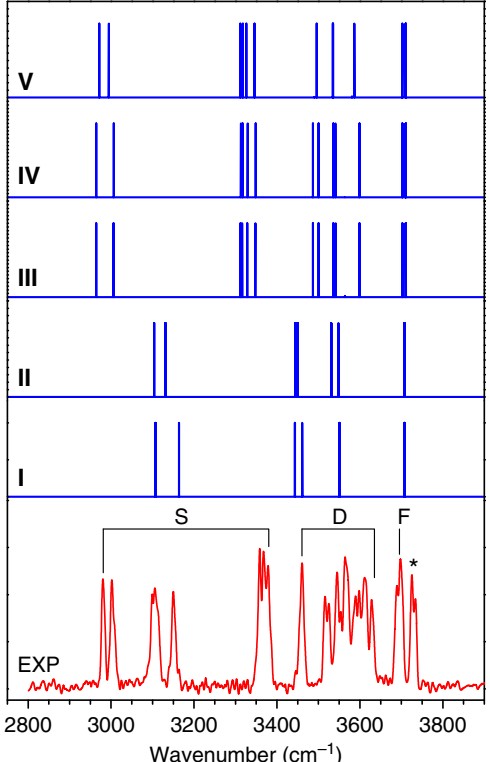

**Fig. 1 Comparison of experimental and simulated IR spectra of the water octamer.** The OH stretch fundamentals assigned to H-donor-free OH (F), double H-donor OH stretch (D), and single H-donor OH stretch (S) are labeled. The simulated spectra of isomers **I** to **V** are also shown. The calculations were performed at the ab initio MP2/aug-cc-pVDZ level, with the harmonic frequencies scaled by 0.956.

**Fig. 2 Optimized structures of coexisting isomers of ($H_2O$)$_8$ (O, red; H, light gray).** Relative energies from MP2/AVDZ and DLPNO-CCSD(T)/AVTZ (in parenthesis) are listed in kcal/mol. Point group symmetries of the isomers are noted in parenthesis.

primarily in the orientation of hydrogen bonds within the distorted cubes.

As pointed out previously[27,30], direct comparison between theory and experiment for the relative intensities of vibrational bands is very difficult, owing to the complexity of experiment (IR absorption combined with dissociation, saturation effects, etc.) as well as the limitation of theoretical calculation (neglection of intermolecular zero-point motions). Here, the stick spectra of calculated harmonic vibrational frequencies are utilized to compare with the experimental data. Figure 1 shows the comparison of experimental spectrum of the water octamer and calculated spectra of isomers **I**–**V**. The harmonic OH stretch vibrational frequencies of isomers **I**–**V** are listed in Supplementary Tables 2–5 and the animation of vibrational modes responsible for the experimental bands is given in Supplementary Data 1.

Each of the isomers **I**–**V** possesses three types of OH groups, namely, the OH of water with single hydrogen-donor configuration (single H-donor OH), double H-donor OH, and H-donor-free OH groups. As noted previously[11,24,27–29], the AAD → ADD hydrogen bonds are remarkably shorter than ADD → AAD hydrogen bonds and the corresponding frequency of single H-donor OH stretch is typically lower than that of double H-donor OH stretch (vide infra). Due to the high symmetry of the cubic structures, the normal modes of vibrational stretch of a given type of OH group differ from the other type. As a result, the vibrational frequencies of the single H-donor OH, double H-donor OH, and H-donor-free OH groups are well separated in the OH stretch spectra (Fig. 1 and Supplementary Tables 2–5).

In the calculated spectrum of isomer **I** ($D_{2d}$) (Fig. 1, trace **I**), the band positions of single H-donor OH stretches (3107 and 3164 cm$^{-1}$, Supplementary Table 2) are consistent with the experimental values (3106 and 3150 cm$^{-1}$, Supplementary Table 1); the calculated transitions at 3443 and 3461 cm$^{-1}$ are attributed to the double H-donor symmetric OH stretches (D$^{sym}$) and agree with the experimental absorption centered at 3460 cm$^{-1}$; the calculated band at 3551 cm$^{-1}$ is due to the double H-donor antisymmetric OH stretches (D$^{asym}$) and falls in the experimentally spectral range of 3526–3628 cm$^{-1}$; the calculated H-donor-free OH stretches (3708 cm$^{-1}$) agree well with the experimental value of 3698 cm$^{-1}$. The calculated IR spectrum of isomer **II** ($S_4$) is rather similar to that of isomer **I** ($D_{2d}$) due to similar geometries. In the isomers **I** and **II**, the most significant spectral difference is found in the single H-donor OH stretch region. The two single H-donor OH stretches in isomer **II** are predicted at 3104 and 3131 cm$^{-1}$ with a separation of 27 cm$^{-1}$ (Supplementary Table 3), which might be responsible for the broad band observed at 3106 cm$^{-1}$. The calculated IR spectra of isomers **I** and **II** are much too simple to explain the observed absorptions at 2980, 3002, and 3378 cm$^{-1}$, but these features match rather well with those of isomers **III**, **IV**, and **V** (Fig. 1) that are energetically low-lying. Moreover, the **III**, **IV**, and **V** isomers yield various double H-donor OH stretch vibrational fundamentals that cover the spectral range of 3487–3599 cm$^{-1}$ (Supplementary Tables 4 and 5), which are consistent with the

experimentally congested bands in the 3516–3628 cm$^{-1}$ region. The agreement of the calculated spectra with experiment is reasonable to confirm the assignment of the **I**–**V** isomers responsible for the experimental spectra.

In addition, the two well-separated free OH bands at 3698 cm$^{-1}$ (labeled F) and 3726 cm$^{-1}$ (marked with an asterisk) can be related to two distinct AD and AAD sites, because the H-donor free OH groups of the AAD sites generally appear at ~3700 cm$^{-1}$ and those of the AD sites at a higher-frequency range[6,9]. The asterisk-labeled band likely originates from a non-cubic isomer of water-solvated heptamer (Supplementary Fig. 3).

The five isomers **I**–**V** all have interesting cubic structures. The fact that the five cubic isomers **I**–**V** lie within 3 kcal/mol indicates that they can possibly coexist at the finite temperature of experimental condition. The interconversion barrier among them is larger than 4 kcal/mol at the MP2/AVDZ level (Supplementary Fig. 4). For instance, the interconversion between the two enantiomeric isomers **III** and **IV** need go through four transition states and three intermediates, with the largest barrier of about 5 kcal/mol. Such interconversion barrier might be sufficiently large so that the ultra-high-pressure supersonic expansion cooling is capable of kinetically quenching the non-equilibrium octamer system prior to its rearrangement to the global-minimum-energy structure[22]. Quenching in our experiment produces a non-equilibrium distribution, which benefits to the observation of all five cubic isomers. To evaluate the temperature effect on the distribution of the isomers, Gibbs free energies $\Delta G$ of isomers **I**–**V** were calculated for the temperature from 0 to 300 K (Supplementary Fig. 5). Clearly, the free energy difference $\Delta G_{II-I}$, $\Delta G_{III-I}$, $\Delta G_{IV-I}$, and $\Delta G_{V-I}$ does not alter significantly below room temperature, indicating that the population of the five isomers changes little at low temperature.

**Analysis of the electronic structure.** To understand the electronic structure of the water octamer, we have analyzed the hydrogen-bond (HB) network of the cubic isomers using delocalized and localized molecular orbital (MO) theory. Theoretical approaches were applied of natural bond orbital (NBO)[36], adaptive natural density partitioning (AdNDP)[37], energy decomposition analysis–natural orbitals for chemical valence (EDA-NOCV)[38], and principal interacting orbital (PIO) analysis[39]. Hydrogen bonding between an O–H antibonding orbital (denoted σ*(O–H)) and an adjacent oxygen lone-pair donor can be viewed as a three-center two-electron (3c–2e) interaction, which features the O lone-pair delocalizing to the H–O antibonding region (Supplementary Fig. 6)[15,40]. As exemplified by water dimer, the contribution of 3c–2e HB energy to the intrinsic total binding energy ($E_{HB}/E_{total}$) is about 81.4% from EDA-NOCV analysis, whereas the PIO contribution from the interaction between the lone pair and the σ*(O–H) antibond is about 88.7% for each 3c–2e HB (Supplementary Fig. 6). As shown by the bond distances (Supplementary Tables 7–11), bond orders, and hybrid orbitals (Supplementary Tables 12–16), the bond strength of OH groups follows the order of the single H-donor

O–H < double H-donor O–H < H-donor-free O–H, which mirrors the extent of electron donation from O lone pair to the σ (OH)* antibonding orbitals and accounts for the sequence of the corresponding OH stretch vibrational frequencies observed.

For the water octamer, the $E_{HB}/E_{total}$ values of isomers **I–V** are all around 89% (Supplementary Table 6), which are considerably larger than that in the water dimer (81%). This enhanced HB interaction can be partially attributed to the extensively delocalized HB network (vide infra). In isomer **I** ($D_{2d}$), the AAD → ADD hydrogen bonds (1.698 Å) are much shorter than ADD → AAD hydrogen bonds (1.904 Å) (Supplementary Table 7). The NBO second-order perturbation energy ($E_2$) analysis of the $D_{2d}$ isomer **I** (Supplementary Table 7) shows that the AAD → ADD interaction energies (e.g., $E_2(O^1 \cdots H–O^5) = 32.17$ kcal/mol) are remarkably larger than the ADD → AAD interaction energies (e.g., $E_2(O^1H \cdots O^4) = 12.64$ kcal/mol), indicating that the face-to-face stacking of the two tetramer rings ($O^1–O^2–O^3–O^4$ and $O^5–O^6–O^7–O^8$) is highly favorable. The significantly strong AAD → ADD interactions are also found in the **II–V** isomers (Supplementary Tables 8–11) and benefit the formation of water cubes as well as the stacking of cubic and hexagonal layers that occur in the condensed phase[23,25].

The five water octamer isomers adopting pseudo-cubic structures are interesting. As each O–H⋯O HB is dominated by the 3c–2e interaction from O lone-pair delocalizing onto the H–O antibonding region, the pseudo-cubic structure can be viewed as consisting of one pair of electron between every two apex oxygen atoms. Interestingly, this bonding pattern is akin to that in the famous cubane ($O_h$-$C_8H_8$)[41], where each C–C bond contains two localized electrons, as shown in Fig. 3. While the cubane structure lies much higher in energy than its ring isomer, the $D_{2d}$ cubic isomer of $(H_2O)_8$ lies much lower in energy than the ring isomer, by 11.64 kcal/mol at the ab initio DLPNO-CCSD(T)/AVTZ level. Consistent with the extensively delocalized HB interaction, the cubic isomer of water has remarkable thermodynamic stability.

Especially noteworthy is the finding that the **III** and **IV** structures among the five isomers **I–V** are rare chiral isomers with $C_2$ symmetry. It is thus interesting to speculate the existence of such transient local chiral structures in bulk water. Previous far-IR vibration–rotation tunneling spectroscopy of chiral cyclic water trimers indicates that rapid quantum tunneling occurs between the enantiomers[10]. Low-temperature scanning tunneling

microscopy shows concerted tunneling of four protons within chiral cyclic water tetramers supported on an inert surface[42]. The calculated vibrational circular dichroism (VCD) and electronic circular dichroism (ECD) spectra of the two chiral water octamers (isomers **III** and **IV**) (Supplementary Figs. 7 and 8) show clear chiral recognition peaks and provide incentives for future experimental studies.

It is interesting to note that phase transitions between solid and liquid water have been observed in simulations of water clusters as small as the octamer, which is supported by the calculated free energy as a function of temperature[19–22]. The present study has identified the coexistence of five water octamer cubes that are stabilized by extensive delocalized HB interaction. These findings provide crucial information for fundamental understanding of the processes of cloud, aerosol, and ice formation, especially under rapid cooling[43–45]. It is hoped that the present results will both provide a benchmark for accurate description of the water intermolecular potentials to understand the macroscopic properties of water and stimulate further study of intermediate-ice structures formed in the crystallization process of ice.

## Methods

**Experimental method**. Experiments were performed using a VUV-FEL-based IR spectroscopy apparatus at the DCLS facility[32]. The DCLS facility delivered the VUV-FEL light with a continuously tunable wavelength region between 50 and 150 nm. The VUV-FEL was operated in the high gain harmonic generation mode[46], in which the seed laser was injected to interact with the electron beam in the modulator. With proper optimization of the LINAC (linear accelerator), a high-quality accelerated electron beam with a beam emittance of ~1.5 mm mrad, an energy spread of ~1% and a pulse duration of ~1.5 ps were obtained. The VUV-FEL pulse is currently operated at 20 Hz, and the maximum pulse energy output is ~500 μJ/pulse (~3 × 10$^{14}$ photons/pulse). For recording the pulse spectral characteristic, an online VUV spectrometer was used to monitor each single VUV-FEL pulse.

Neutral water clusters were produced by supersonic expansions of water seeded in helium using a high-pressure pulsed valve (Even-Lavie valve, EL-7-2011-HT-HRR) that is capable of producing very cold molecular beam conditions[33]. In order to avoid condensation, the operating temperature of the valve and the entire gas inlet was 353 K. The molecular beam passed through a 4 mm diameter skimmer and an aperture with 3 mm opening. The extraction plates of reflectron time-of-flight mass spectrometer (TOF-MS) were powered by a high-voltage direct current (DC) of 2950 V. Charged clusters were deflected out of the molecular beam by the DC electric field of the extraction plates. Neutral water clusters in the beam were then near-threshold ionized by the VUV-FEL pulse and mass-analyzed in the reflectron TOF-MS. The tunable IR laser pulse from an OPO was introduced at about 30 ns prior to the VUV-FEL pulse in the same VUV-FEL interaction region. VUV wavelength, pulse energy, and beam conditions were optimized to maximize

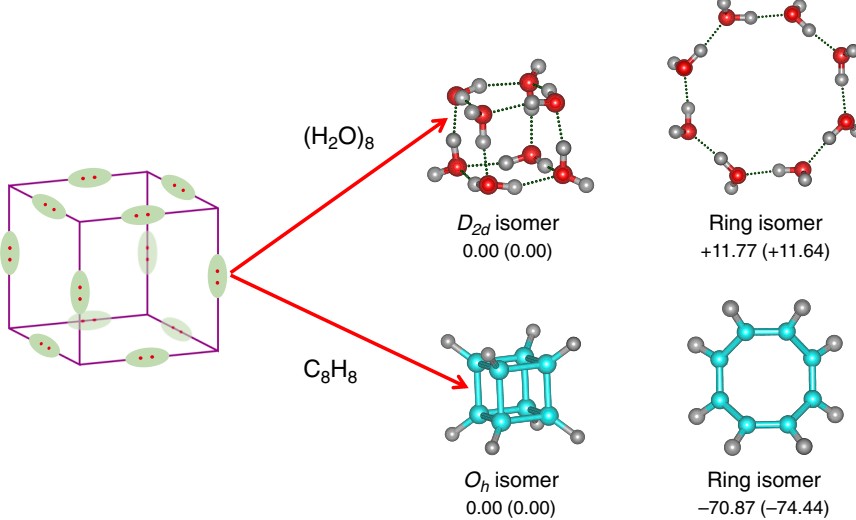

$(H_2O)_8$

$C_8H_8$

$D_{2d}$ isomer
0.00 (0.00)

Ring isomer
+11.77 (+11.64)

$O_h$ isomer
0.00 (0.00)

Ring isomer
−70.87 (−74.44)

**Fig. 3 Analog of the isomers for $(H_2O)_8$ and $C_8H_8$ (O, red; H, light gray).** Relative energies from MP2/AVDZ and DLPNO-CCSD(T)/AVTZ (in parenthesis) are listed in kcal/mol. The pseudo-cubic $D_{2d}$ structure of $(H_2O)_8$ can be viewed as consisting of one pair of electron between every two apex oxygen atoms, which bonding pattern is analogous to that in the $O_h$ structure of cubane ($O_h$-$C_8H_8$) with each C–C bond containing two localized electrons.

the signal of a size-specific water cluster of interest with no interference from larger clusters. When the resonant vibrational transition is hit by the IR laser light and causes vibrational predissociation, a depletion of the selected neutral cluster mass signal will be detected. At 113.30 nm, the IR spectrum of size-selected neutral water octamer was obtained as a depletion spectrum of the monitored $(H_2O)_8^+$ ion signal for the cluster by scanning the IR wavelength and normalizing to parent ion signal. Typical spectra were recorded by scanning the dissociation laser in steps of 2 cm$^{-1}$ and averaging over 600 laser shots at each wavelength. The VUV-FEL in the present experiment was operated at 20 Hz and IR laser was operated at 10 Hz. IR spectra were recorded in the difference mode of operation (IR laser on–IR laser off). IR spectrum was determined from the relative depletion of the mass spectrometric ion signal ($I_0$ and $I(v)$) and the frequency-dependent laser power $P(v)$ using $\sigma = -\ln[I(v)/I_0]/P(v)$. The normalization with the IR laser pulse energy accounted for its variations over the tuning range. IR power dependence of the signal was measured to ensure that the predissociation yield is linear with photon flux. IR-VUV scheme of neutral $(H_2O)_8$ is free from spectral contamination due to the fact the IR excited water clusters dominantly dissociate into the monomer and protonated cluster cation mass channels in the VUV photoionization process[47].

The tunable IR laser beam was generated by a KTP/KTA OPO/OPA system (LaserVision) pumped by an injection-seeded Nd:YAG laser (Continuum Surelite EX). This system was tunable from 700 to 7000 cm$^{-1}$ with a line width of 1 cm$^{-1}$. The wavelength of the OPO laser output was calibrated using a commercial wavelength meter (HighFinesse GmbH, WS6-200 VIS IR).

**Structure, energy, and vibrational frequency calculations**. Global-minimum structure search based on DFT was carried out for $(H_2O)_8$ using TGMin code[34,35,48]. Totally 2784 structures have been found. Quantum chemical calculations were carried out to refine the energies of the low-lying isomers (within 11 kcal/mol) at the MP2/aug-cc-pVDZ (AVDZ) level of theory using the Gaussian 09 package[49]. Harmonic vibrational frequencies were calculated with analytical second derivatives of energy. A scaled factor of 0.956 was used for harmonic vibrational frequencies to account for the systematic errors in the calculations[50–52]. The MP2/AVDZ relative energies and energy barriers were calculated at 0 K with zero-point vibrational energies. The DLPNO-CCSD(T)/aug-cc-pVTZ (AVTZ) relative energies were calculated at the MP2/AVDZ optimized geometries with the ORCA program[53–55], which included the MP2/AVDZ zero-point vibrational energy corrections. The free energies $G(T) = U(T) + PV − TS$ of these low-lying isomers were calculated using the MP2/AVDZ method for the vibrational analyses ($U$, $S$, $P$, $V$, and $T$ stand for the internal energy, entropy, pressure, volume, and temperature, respectively). The VCD and ECD spectra of two chiral water octamers (isomers **III** and **IV**) were calculated at the PBE/TZ2P level of theory[56,57].

**Hydrogen-bonding analysis**. To understand the structure and stability of these water clusters, the hydrogen bonding interaction was analyzed using MO theory and the AdNDP method[37] at the level of MP2/AVDZ. The AdNDP analyses yield both localized and semi-localized multi-center bonds, providing a chemically intuitive bonding picture for complicated molecular systems, especially those with extensive electron delocalization.

The AdNDP bonding analyses demonstrate there is $n$ three-center two-electron (3c–2e) hydrogen bonding interaction in each structure of water octamer. The remaining ones are one-center two-electron (1c–2e) lone pairs, two-center two-electron (2c–2e) O–H σ bonds, and the O–H bonds along the hydrogen bond axis.

The nature of the hydrogen bonding interaction in the cubic isomers was further analyzed with the EDA-NOCV analyses[38], at the level of GGA PBE/TZ2P[58] using the Amsterdam Density Functional program (ADF 2016.101)[59,60]. The EDA-NOCV scheme provides both qualitative ($\Delta\rho_{orb}$) and quantitative ($\Delta E_{orb}$) information about the strength and contribution of orbital interactions in chemical bonding. We used the unrelaxed water fragments from the optimized water cluster structures to derive the intrinsic binding energies of waters in cluster.

## Data availability
The data that support the findings of this study are available from the corresponding author upon reasonable request. Source data are provided with this paper.

## Code availability
The code that support the findings of this study are available from the corresponding author upon reasonable request.

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

## Acknowledgements
We gratefully acknowledge Mark A. Johnson for helpful discussions. The authors thank the DCLS for VUV-FEL beam time and the DCLS staff for support and assistance. This work was supported by the National Natural Science Foundation of China (21688102, 21673231, 21590792. and 91645203), the Strategic Priority Research Program of Chinese Academy of Sciences (CAS) (XDB17000000), International Partnership Program of CAS (121421KYSB20170012), Dalian Institute of Chemical Physics (DICP DCLS201701 and DCLS201702), CAS (GJJSTD20190002), the Science Challenge Project (TZ2016004), and Guangdong Key Laboratory of Catalytic Chemistry. The calculations were performed on SUSTech supercomputer.

## Author contributions
L.J., J.L., and X.Y. designed the research. G.L., Q.L., C.W., Y.Y., B.Z., W.Z., D.D., G.W., L.J., and X.Y. performed the experiments and data analysis. Y.Y.Z., H.S.H., D.H.Z., and J.L. performed the theoretical calculations and data analysis. L.J., J.L., and X.Y. cowrote the manuscript.

## Competing interests
The authors declare no competing interests.
