## [Peer Review File · Nature Communications]

REVIEWER COMMENTS

Reviewer #1 (Remarks to the Author):

This paper reports experimental vibrational spectra of the gas-phase (H₂O)₈ cluster, showing conclusively, that contributions from the low-lying D_{2d} and S₄ as well as higher energy cubic structures. The results will be of considerable interest to researchers working on hydrogen-bonding in water networks. However, revision is required before publication is possible. In particular, I believe that the authors have to downplay the hype. Specific issues are listed below.

1. Page 3. I don't see why the authors refer to the structures of the hexamer and heptamer as "complex"

2. Page 3. Right after mentioning the octamer, the authors state "Experiments strongly suggest the presence of ice nanocrystals". What is the connection between ice nanocrystals and the octamer?

3. Page 3. It is not clear what is meant by "diverse component"

4. Page 4. The authors state "Multiple coexisting cubic octamers provide a coherent picture of structural diversity of bulk water and a cluster-scale precursor to the phase transition between solid and liquid water." The cubic isomers say nothing about the diversity of liquid water, and the "solid-liquid" transition in (H₂O)₈ is very different from that in liquid water.

5. Page 6. It is not clear what the authors mean by "implicit description of intermolecular ZPE". MP2 or DFT calculations of vibrational frequencies do not assume an implicit description of ZPE.

6. Page 6, "Each structure of isomers I–V possesses" should be reworded as "Each of the isomers I–V possesses".

7. Page 7 "analogous geometries" should be reworded "similar geometries"

8. Page 7. The authors state "Under the pulsed supersonic expansion condition in the present work, the presence of all five cubic isomers is quite surprising, indicating that our VUV-FEL spectroscopic technique is apt to explore low-lying neutral isomers unknown before". Why is this surprising that these higher energy isomers are seen? This is common in expansion studies of clusters. Also, these isomers were not unknown before. All of them are discussed in the paper of Tsai and Jordan.

9. Page 7, Last paragraph. The sentence about the population of higher lying isomers and the Boltzmann distribution, by itself is not useful. One also needs to know the temperature. For cold, equilibrium clusters one would not see these higher lying isomers. The authors must be seeing these because quenching in their experiment produces a non-equilibrium distribution.

10. Page 8. It makes no sense to report distributions up to 1000K at temperatures above about 300K, the clusters will evaporate on the time scale of the experiment.

11. Page 9. I believe that it is stretching things to call the bonding in these clusters aromatic.

12. Page 9. The authors state "The five water octamer isomers adopting pseudo-cubic structure is highly remarkable". I don't see what is remarkable about this. The cubic structures maximize the number of H bonds. Plus, as noted above, all five of these structures were predicted in theoretical studies.

13. Page 10. The authors again refer to the unexpected coexistence of several isomers. Again, I do not believe this is unexpected.

14. Page 11. The authors conclude with a reference to crystallization of ice. But they don't make a case that the octamer is relevant to such crystallization.

Reviewer #2 (Remarks to the Author):

This paper reports an important investigation of a water cluster of central importance in untangling details of the solid to liquid transition in water. The use of a sophisticated new technology reveals a

much higher degree of complexity in the energetically available isomer distribution than previous infrared spectroscopic studies of the octamer have shown. The accompanying theoretical calculations permit a reasonably credible assignment of the observed spectral features to particular isomers. This work is likely to stimulate further investigations into the remarkably complicated nature of the octamer.

Both the experimental and theoretical work seems to have been done at a high level, and the paper is written in an interesting fashion(although it could benefit from some preening of the English). I am thus pleased to recommend in favor of publishing this paper, essentially without change, except for the above comment(at the option of the authors), and the addition of two new references:

An updated reference on the study of water clusters should appear as Ref 4:

F.N. Keutsch, and R.J. Saykally, "Water Clusters: Untangling the mysteries of the liquid, one molecule at a time," PNAS 98, 10533-10540 (2001).

Another study of the octamer by terahertz spectroscopy should appear as Ref 31:

Richardson, J.O., Wales, D.J., Althorpe, S. C., McLaughlin, Shih, O. and Saykally, R.J. "Investigation of Terahertz Vibration–Rotation Tunneling Spectra for the Water Octamer" J. Phys. Chem. A. 117 (32), 6960-6966 (2013)

Response Letter

- Title: **Infrared spectroscopic study of hydrogen bonding topologies in the water octamer: The smallest ice cube**
- Tracking #: NCOMMS-20-31301

We are very grateful to the critical comments and constructive suggestions provided by the two reviewers, which have significantly helped us to improve the manuscript. Our manuscript has been revised accordingly, and the changes are highlighted by blue color in the text. The following lists our responses (in blue color) to the comments from the two reviewers.

Reviewer #1:

Reviewer #1 (Remarks to the Author):

This paper reports experimental vibrational spectra of the gas-phase (H₂O)₈ cluster, showing conclusively, that contributions from the low-lying D_{2d} and S₄ as well as higher energy cubic structures. The results will be of considerable interest to researchers working on hydrogen-bonding in water networks. However, revision is required before publication is possible. In particular, I believe that the authors have to downplay the hype. Specific issues are listed below.

Response: Thanks for the critical comments and constructive suggestions. We have revised the manuscript following your suggestions.

1. Page 3. I don't see why the authors refer to the structures of the hexamer and heptamer as "complex".

Response: We agree with the referee that while water is a chemical complex, the water cluster is better not called a “complex”. Therefore, the "complex" word has been deleted.

2. Page 3. Right after mentioning the octamer, the authors state *"Experiments strongly suggest the presence of ice nanocrystals"*. What is the connection between ice nanocrystals and the octamer?

Response: Our statement was based on the following: The low-energy structures of the water octamer were predicted to be nominally cubic, with the eight tri-coordinated water molecules taking up positions at the corners of the cube. Such tri-coordinated water molecules have been identified at the surface of ice.

Given the referee’s comment, we replaced "Experiments strongly suggest the presence of ice nanocrystals" by the more detailed description in the revised manuscript.

3. Page 3. It is not clear what is meant by *"diverse component"*

Response: The "diverse component" has been changed to "disordered component".

4. Page 4. The authors state *"Multiple coexisting cubic octamers provide a coherent picture of structural diversity of bulk water and a cluster-scale precursor to the phase transition between solid and liquid water."* The cubic isomers say nothing about the diversity of liquid water, and the *"solid-liquid"* transition in $(H_2O)_8$ is very different from that in liquid water.

Response: We agree that the original statement is inaccurate. Therefore, the sentence "Multiple coexisting cubic octamers provide a coherent picture of structural diversity of bulk water and a cluster-scale precursor to the phase transition between solid and liquid water" has been removed from the manuscript.

5. Page 6. It is not clear what the authors mean by *"implicit description of intermolecular ZPE"*. MP2 or DFT calculations of vibrational frequencies do not assume an implicit description of ZPE.

Response: The referee is certainly correct; our sentence is confusing, and "implicit description of intermolecular zero-point motions" has now been changed to "neglect of intermolecular zero-point motions".

6. Page 6, *"Each structure of isomers I–V possesses"* should be reworded as *"Each of the isomers I–V possesses"*.

Response: The "Each structure of isomers I–V possesses" has been changed to "Each of the isomers I–V possesses". Thanks!

7. Page 7 *"analogous geometries"* should be reworded *"similar geometries"*

Response: The "analogous geometries" has been changed to "similar geometries".

8. Page 7. The authors state *"Under the pulsed supersonic expansion condition in the present work, the presence of all five cubic isomers is quite surprising, indicating that our VUV-FEL spectroscopic technique is apt to explore low-lying neutral isomers unknown before"*. Why is this surprising that these higher energy isomers are seen? This is common in expansion studies of clusters. Also, these isomers were not unknown before. All of them are discussed in the paper of Tsai and Jordan.

Response: Sorry for missing noting the reference. Given this fact, the sentence "Under the pulsed supersonic expansion condition in the present work, the presence of all five cubic isomers is quite surprising, indicating that our VUV-FEL spectroscopic technique is apt to explore low-lying neutral isomers unknown before" has been removed from the manuscript.

9. Page 7, Last paragraph. The sentence about the population of higher lying isomers and the Boltzmann distribution, by itself is not useful. One also needs to know the temperature. For cold, equilibrium clusters one would not see these higher lying isomers. The authors must be seeing these because quenching in their experiment produces a non-equilibrium distribution.

Response: We agree with the reviewer's comment. Quenching in our experiment produces most likely a non-equilibrium distribution, which benefits to the observation of all five cubic isomers. Such discussion has been added into the manuscript.

10. Page 8. It makes no sense to report distributions up to 1000 K at temperatures above about 300K, the clusters will evaporate on the time scale of the experiment.

Response: Thanks for the comments. The temperature range has been changed to be 0–300 K in the text and figure.

11. Page 9. I believe that it is stretching things to call the bonding in these clusters aromatic.

Response: Thanks for the comments. We have removed this part about the bonding in these clusters being aromatic.

12. Page 9. The authors state "The five water octamer isomers adopting pseudo-cubic structure is highly remarkable". I don't see what is remarkable about this. The cubic structures maximize the number of H bonds. Plus, as noted above, all five of these structures were predicted in theoretical studies.

Response: Thanks for the comments. We intended to say that a non-planar cubic structure looks remarkable. Indeed it is a result of maximizing the number H-bonds, as is predicted theoretically. We therefore revised this sentence.

13. Page 10. The authors again refer to the unexpected coexistence of several isomers. Again, I do not believe this is unexpected.

Response: Thanks for the comments. The "unexpected" word has been removed from the manuscript.

14. Page 11. The authors conclude with a reference to crystallization of ice. But they don't make a case that the octamer is relevant to such crystallization.

Response: Thanks for the comments. Nanometer-sized water clusters are engaged in a range of processes of cloud, aerosol, and ice formation, especially under rapid cooling (Ref. 42-44). We agree that it is hard to make a clear case that the octamer is directly relevant to crystallization of ice. Nevertheless, our observation of five water octamer cubes suggest that it cannot be fully excluded that there may exist a family of microcrystalline structures in ice formation that is yet to be fully recognized.

Reviewer #2:

This paper reports an important investigation of a water cluster of central importance in untangling details of the solid to liquid transition in water. The use of a sophisticated new technology reveals a much higher degree of complexity in the energetically available isomer distribution than previous infrared spectroscopic studies of the octamer have shown. The accompanying theoretical calculations permit a reasonably credible assignment of the observed spectral features to particular isomers. This work is likely to stimulate further investigations into the remarkably complicated nature of the octamer.

Both the experimental and theoretical work seems to have been done at a high level, and the paper is written in an interesting fashion (although it could benefit from some preening of the English). I am thus pleased to recommend in favor of publishing this paper, essentially without change, except for the above comment (at the option of the authors), and the addition of two new references:

An updated reference on the study of water clusters should appear as Ref 4:

F.N. Keutsch, and R.J. Saykally, "Water Clusters: Untangling the mysteries of the liquid, one molecule at a time," PNAS 98, 10533-10540 (2001).

Another study of the octamer by terahertz spectroscopy should appear as Ref 31:

Richardson, J.O., Wales, D.J., Althorpe, S. C., McLaughlin, Shih, O. and Saykally, R.J. "Investigation of Terahertz Vibration–Rotation Tunneling Spectra for the Water Octamer" J. Phys. Chem. A. 117 (32), 6960-6966 (2013).

Response: We thank the reviewer very much for the delightful comments. We have also tried our best to preen the English in the revised version.

The reference “F.N. Keutsch, and R.J. Saykally, Water Clusters: Untangling the mysteries of the liquid, one molecule at a time. PNAS 98, 10533-10540 (2001)” has been updated as Ref. 4 as suggested by the reviewer.

The reference “Richardson, J.O., Wales, D.J., Althorpe, S. C., McLaughlin, Shih, O. and Saykally, R.J. Investigation of Terahertz Vibration–Rotation Tunneling Spectra for the Water Octamer. J. Phys. Chem. A. 117 (32), 6960-6966 (2013)” has been cited as Ref. 31 as suggested by the reviewer.

REVIEWERS' COMMENTS

Reviewer #1 (Remarks to the Author):

The authors did a satisfactory job at addressing the issues raised in my earlier review. I believe the paper is now suitable for publication in Nature Comm. I believe the paper will attract considerable attention from both experimental and theoretical chemists.

Response Letter

- Title: **Infrared spectroscopic study of hydrogen bonding topologies in the water octamer: The smallest ice cube**
- Tracking #: NCOMMS-20-31301A

REVIEWERS' COMMENTS

Reviewer #1 (Remarks to the Author):

The authors did a satisfactory job at addressing the issues raised in my earlier review. I believe the paper is now suitable for publication in Nature Comm. I believe the paper will attract considerable attention from both experimental and theoretical chemists.

Author Response: We thank the reviewer very much for the delightful comments.